# Eighteen-Month Orthodontic Bracket Survival Rate with the Conventional Bonding Technique versus RMGIC and V-Prep: A Split-Mouth RCT

Victor Ghoubril [1,*], Joseph Ghoubril [1], Maher Abboud [2], Tatiana Bou Sakr [3], Louis Hardan [4,*] and Elie Khoury [1]

1 Department of Orthodontics, School of Dentistry, Saint-Joseph University, Beirut 1107 2180, Lebanon; joseph.ghoubril@usj.edu.lb (J.G.); elie.khoury@usj.edu.lb (E.K.)
2 Unité Environnement Génomique et Protéomique, U-EGP, Faculté des Sciences, Saint-Joseph University, Campus des Sciences et Technologies, Mar Roukos-B.P. 1514, Riad El Solh, Beirut 1107 2050, Lebanon; maher.abboud@usj.edu.lb
3 Department of Removable Prosthodontics, School of Dentistry, Saint-Joseph University, Beirut 1107 2180, Lebanon; tatiana.bousakr@net.usj.edu.lb
4 Department of Restorative Dentistry, School of Dentistry, Saint-Joseph University, Beirut 1107 2180, Lebanon
* Correspondence: victor.ghoubril1@usj.edu.lb (V.G.); louis.hardan@usj.edu.lb (L.H.); Tel.: +961-70601038 (V.G.)

**Abstract:** The association of the V-prep and a resin-modified glass ionomer cement (RMGIC) has shown to be a suitable alternative for the orthodontic bracket bonding procedure in vitro. The aim of this study was to evaluate over eighteen months the clinical bonding failure and survival rates of the conventional bonding technique using the Transbond XT (3M Unitek, Monrovia, CA, USA) and the RMGIC Fuji Ortho LC (GC Corporation, Tokyo, Japan) prepared with the V-prep. Therefore, one operator using the straight-wire technique bonded two hundred metallic brackets to upper and lower premolars of twenty-five patients requiring an orthodontic treatment. The randomized trial was a single-blind design in a split-mouth comparison. Each patient was randomly allocated one of the two bonding systems for each premolar on each side of the mouth. The bonding and rebonding techniques were standardized throughout the trial and bond failure was recorded each month for a period of eighteen months. The survival rates of the brackets were estimated by Kaplan–Meier and log-rank test ($p < 0.05$). A total of 200 orthodontic brackets were included in the study with a significant lower failure rate of 9.0% for the V-prep and RMGIC compared to 25.0% for the conventional bonding technique ($p < 0.05$). A higher survival rate was observed for the V-prep and RMGIC (16.36 months) over the conventional bonding technique (13.95 months) ($p < 0.05$). Lower premolar bonding failure was higher than upper premolar for both bonding techniques. The V-prep followed by RMGIC, with enamel surface protection abilities, can be used as an alternative bonding technique in an orthodontic treatment.

**Keywords:** resin-modified glass ionomer; surface preparation; orthodontic brackets; V-prep; dental acid etchants; phosphoric acid; surface protection; survival rate

## 1. Introduction

A successful orthodontic treatment should achieve the expected outcomes within an appropriate length of time [1]. Therefore, the increasing demand for orthodontic treatment keeps orthodontists chasing the optimal bonding strategy considering different brackets and surfaces [2]. At the beginning, Zachrisson introduced the 37% orthophosphoric acid for enamel surface preparation and the conventional composite resin bonding technique for a metallic orthodontic [3]. The transbond XT (3M Unitek, Monrovia, CA, USA) has been used as a two-stage standard bonding material for most comparison studies [4–6]. Self-etch primers (SEPs) have been introduced to be a one-stage bonding technique with comparable

shear bond strength (SBS) results [7]. To decrease the bonding procedure chair-time, the pre-coated system, APC Flash-Free Adhesive Coated Appliance System developed by 3M™ Unitek (Monrovia, CA, USA), has also been tested with comparable bond strength to the conventional technique [8]. Despite all the advantages, composite resins impose the risk of demineralization of enamel adjacent to brackets and requires a dry surface [9]. The resin modified glass ionomer cement (RMGIC) releases fluoride and prevents enamel demineralization [10]. However, studies have shown significantly lower bond strengths of the RMGIC when compared to the resin composite in orthodontics [11,12]. The RMGIC Fuji Ortho LC (GC Corporation, Tokyo, Japan) has shown upgraded bond strength when used on etched and dried enamel [13]. Still, the results with the conventional acid etch and RMGIC were not satisfying when compared to the acid etch and the composite resin [14]. Sandblasting and enamel deproteinization can improve SBS of the RMGIC [15]. A previous in vitro study, published by the same authors, introduced the V-prep as a mixture of sodium hypochlorite and acid etch to prepare the tooth enamel surface by cleaning the organic layer and enhancing the chemical and mechanical bonding of the RMGIC. This study showed comparable results with the conventional composite resin when the enamel surface was prepared with the V-prep and bonded with the Fuji Ortho LC (GC Corporation, Tokyo, Japan) [16]. Another study showed equal or lower toxicity for the V-prep compared to the conventional acid etch when tested on gingival fibroblasts cells for 30 s [17]. Yet, no clinical trial used the V-prep as a preparing agent before the bonding with RMGIC.

Laboratory in vitro or ex vivo tests and comparisons were performed on multiple adhesive techniques in orthodontics [18–22]. Clinical trials comparing SEP to conventional showed comparable results of bonding failure for both products and were conducted for six months [23], twelve months [24,25] and eighteen months [26]. A systematic review found that ignoring debonding risk factors was the major limitation in comparing articles and studies developing the bracket failure rate of different bonding techniques over eighteen months of clinical trials [27]. Thus, to reduce study biases, it is very important to predict the risk factors associated with bracket bonding failure [28]. The increased overbite showed statistically significant differences in bracket failure rate. Other risk factors such as younger patients, class II patients and mandibular molars and premolars increased the bracket bond failure [29]. Moreover, there was more incidence of bond failure when lighter alignment wires were used in the first six months of treatment [30]. Another bias can be associated with the cleaning and the rebonding procedure of the orthodontic bracket during the treatment affecting the SBS and the survival rate of the bracket [31]. The Er:YAG laser procedure could be considered an optimal method for promoting the rebonding orthodontic brackets while direct flame, mechanical grinding or sandblasting obtained clinically acceptable bond strength values [32]. In-office methods, slow speed carbide bur and ultrasonic scaler are effective, quick and cheap methods for bracket base cleaning for rebonding with slight lower values when compared to a replaced new bracket [33]. The use of a magnification system while cleaning the tooth enamel before rebonding a bracket can also affect the surface properties [34,35]. Also, the use of tungsten carbide bur requires multistep polishing to minimize enamel loss [36]. RMGIC can be safely cleaned with tungsten carbide burs while conventional composite resin and SEPs require multi-step polishing [37].

To the authors knowledge, bracket bonding survival rate RCTs lacked standardizations regarding clear bonding and rebonding procedures. No clinical trial compared the new product V-prep with RMGIC to the conventional bonding technique. Therefore, the aim of this randomized clinical trial is to compare the survival rate of a conventional bracket bonding procedure to the V-prep and RMGIC bonding procedure on a split-mouth clinical study, using a clear protocol for bonding, rebonding and tooth cleaning, and considering all the risk factors to eliminate any bias. The null hypothesis to be assessed is that the conventional bonding technique and the V-prep followed by RMGIC technique would provide similar bond failure and survival rates.

## 2. Materials and Methods

### 2.1. Ethical Approval and Participants

After the approval of this study by the ethical committee of Saint-Joseph University of Beirut (USJ-2020-010), all subjects eligible for inclusion were interviewed in the presence of their parents. After outlining the purpose of the trial, the parents signed the consent form. All selected patients were young school students (12–15 years old) and did not have any restorations in the buccal surfaces of the premolars. They required full orthodontic treatment with bonding on both arches. Crowding was mild or average (less than 5 mm), overbite was mild or average (less than 50%), malocclusion was of class I or mild class II. Patients with required extractions in the treatment plan or with poor oral hygiene were not included in this study. A total of twenty-five patients (13 boys and 12 girls) who met the inclusion criteria were selected. To eliminate the tooth shape and position effect on the bond strength, the sample of the study was restricted to upper and lower first and second premolars [38]. The sample consisted of two hundred premolars' metallic brackets with a 0.022 inch slot Roth prescription (Synergy; Rocky Mountain Orthodontics, Franklin, IN, USA).

### 2.2. Trial Design and Blinding

The trial was registered in the ClinicalTrials.gov (NCT05939102) with a start date (December 2021) and a completion date (June 2023). A split-mouth, cross-quadrant design was used to determine which bonding technique was applied on each premolar. The brackets were bonded so that homologous teeth from the same arch received different materials, and the patient acted as a self-control. A coin was tossed to ensure the randomization of the starting first upper left premolar bonding material in each patient. The patient was not aware which bonding technique (conventional composite resin or V-prep + RMGIC Fuji Ortho LC (GC corporation, Tokyo, Japan) was used on each premolar. It was not possible to blind the operator to the type of bonding material used, as the bonding technique differed between the two systems. To eliminate inter-examiner variation, one operator (VG) performed the bonding procedures.

### 2.3. Sample Size Estimation

Previous studies used a 6% bonding failure rate with the conventional two-stage bonding technique [39,40]. This means a minimum of 87 measurements are needed to have a confidence level of 95% so that the real value is within ± 5% of the measured value. A total of 100 brackets were bonded for each group at the beginning of the treatment.

### 2.4. Bonding Procedure

Before the beginning of the orthodontic treatment, all patients were instructed in oral hygiene and given written instructions about the care of the appliances. The teeth were cleaned with a rubber cup (Addler, Campbelltown, Australia) and water/pumice slurry, rinsed, and isolated using cheek retractors and a low volume suction evacuator. The appropriate bonding technique was applied on each premolar. For the conventional composite resin group, the enamel surface was etched with 37% phosphoric etchant liquid gel (3M Espe, St Paul, MN, USA) for 30 s, rinsed for 15 s and dried for 5 s. A thin uniform coat of primer (Transbond XT Primer; 3M Unitek, Monrovia, CA, USA) was applied. The adhesive resin (Transbond XT Light Cure Adhesive Paste; 3M Unitek, Monrovia, CA, USA) was placed onto the bracket base. The bracket was seated on the enamel surface. The adhesive resin excess was removed with an explorer. Polymerization of the adhesive resin was performed from two directions for a total of 20 s using Ortholux Luminous curing light (3M Unitek, Monrovia, CA, USA). For the V-prep and RMGIC group, the V-prep was applied on the enamel surface for 30 s, rinsed for 15 s and dried for 5 s. The Fuji Ortho LC (GC corporation, Tokyo, Japan) was prepared as described by the manufacturer and placed on the bracket base. The bracket was seated on the enamel surface and the excess of adhesive resin was removed with an explorer. Polymerization of the adhesive resin was

performed from two directions for a total of 20 s using Ortholux Luminous curing light (3M Unitek, Monrovia, CA, USA).

### 2.5. Outcome Measures and Follow-up

All other teeth were bonded using the conventional composite resin technique, and a 0.014 inch NiTi initial aligning archwire (3M Unitek, Monrovia, CA, USA) was tied with elastomeric O-rings. Each patient was monitored every four weeks for eighteen months while undergoing the same treatment maneuvers at the time of the examination, on the right and the left sides. During the periodic examinations, the operator is not aware of the bonding material type used for each premolar. In case of a bond failure, the tooth on which the failure occurred and the date of failure were noted. Only the first bond failure was recorded for each bracket and new bonded brackets were not included in the study. The enamel surface was cleaned with a multi-blade tungsten carbide cutter (H22 Algk.204.016, Komet Dental, Lemgo, Germany) mounted on a red ring contra-angle handpiece (Synea, W&H, Bürmoos, Austria), then, polished with Sof-lex polishing discs (3M Espe, St Paul, MN, USA), and a brand-new replacement bracket was rebounded with the appropriate bonding system and technique as described before. At the end of the treatment, brackets were debonded by gently pressing the wings with a straight orthodontic clip (ETM 800-1001, ATS Plier GmbH & Co. KG, Hasbergen, Germany). The enamel surface was cleaned with a multi-blade tungsten carbide cutter (H22 Algk.204.016, Komet Dental, Lemgo, Germany) mounted on a red ring contra-angle handpiece (Synea, W&H, Bürmoos, Austria), then, polished with Sof-lex polishing discs (3M Espe, St Paul, MN, USA). The clean-up procedure was performed using a $3\times$ magnifying dental loupe (HDL 3.0, Orascoptic, Madison, WI, USA). End-of-treatment pictures were taken, and teeth were examined and compared to the pre-treatment pictures. The appearance of white spot lesions or color differences within the same patient were noted.

### 2.6. Statistical Analysis

A descriptive analysis was first performed: Means and standard deviations (SDs) were used to describe quantitative variables; frequencies and percentages were used to describe qualitative variables. Kaplan–Meier survival analysis was performed to compare the survival rates of conventional bonding technique to that of the V-prep and RMGIC. Statistical significance was determined using a log-rank test. The chi-square test was conducted to assess the potential association between the occurrence of failure and the type of bonding. A *p*-value of less than 0.05 was considered statistically significant. The statistical analysis was conducted using the IBM SPSS statistics software (version 25).

## 3. Results

No dropout patients occurred after eighteen months of treatment. With 100 brackets bonded for each group at the beginning of the treatment, a total of 200 orthodontic brackets (N = 200) were included in the study and they were distributed as follows: 50.0% bonded by conventional method and 50.0% bonded by V-prep + RMGIC; 50.0% were placed on the upper jaw premolars and 50.0% were placed on the lower jaw premolars (Table 1).

**Table 1.** Sample characteristics (N = 200).

| Variable | N | Percentage (%) |
|---|---|---|
| **Bonding product** | | |
| Composite | 100 | 50.0 |
| V-prep + RMGIC | 100 | 50.0 |
| **Jaw** | | |
| Upper | 100 | 50.0 |
| Lower | 100 | 50.0 |

### 3.1. Bracket Removal/Failure and Bonding Method

The results in terms of the proportion of orthodontic brackets detached by bonding method are presented in Table 2. A total of 34 brackets were debonded and a statistically significant difference was observed between the two types of bonding methods with a significantly higher proportion of failure for brackets bonded using the conventional bonding compared to those bonded using V-prep + RMGIC (25.0% vs. 9.0%, *p* = 0.002).

**Table 2.** Bracket removal by bonding method (N = 200).

| Product | Failure | | *p*-Value |
| --- | --- | --- | --- |
| | **Yes** | **No** | |
| | **N (%)** | **N (%)** | |
| **Composite** | 25 (25.0) | 75 (75.0) | 0.002 |
| **V-prep + RMGIC** | 9 (9.0) | 91 (91.0) | |

When per-maxillary analyses were performed (Table 3), a statistically significant difference between the failure rates for the different bonding methods was observed at the level of the mandible, with a higher failure proportion for the conventional method (36.0% vs. 14.0%, *p* = 0.019). No significant difference was found between the two bonding methods in the upper jaw (*p* = 0.143).

**Table 3.** Bracket removal for the different bonding methods and jaws (N = 200).

| Jaw | Product | Failure | | *p*-Value |
| --- | --- | --- | --- | --- |
| | | **Yes** | **No** | |
| | | **N (%)** | **N (%)** | |
| **Upper** | **Composite** | 7 (14.0) | 43 (86.0) | 0.143 |
| | **V-prep + RMGIC** | 2 (4.0) | 48 (96.0) | |
| **Lower** | **Composite** | 18 (36.0) | 32 (64.0) | 0.019 |
| | **V-prep + RMGIC** | 7 (14.0) | 43 (86.0) | |

When per-tooth type analyses were performed (Table 4), no statistically significant differences were found between the tooth type (first or second premolar) bonding failure using the same technique and on the same jaw (*p* > 0.05).

**Table 4.** Bracket removal for the different bonding methods by tooth type (N = 200).

| Jaw | Product | Failure by Tooth Type | | *p*-Value |
| --- | --- | --- | --- | --- |
| | | **1st Premolar** | **2nd Premolar** | |
| | | **N (%)** | **N (%)** | |
| **Upper** | **Composite** | 5 (5.0) | 2 (2.0) | 0.414 |
| | **V-prep + RMGIC** | 0 (0.0) | 2 (2.0) | 0.315 |
| **Lower** | **Composite** | 11 (11.0) | 7 (7.0) | 0.484 |
| | **V-prep + RMGIC** | 3 (3.0) | 4 (4.0) | 0.785 |

### 3.2. Survival Time and Bonding Method

Mean survival time was found to be significantly higher for V-prep + RMGIC compared to conventional bonding method (*p* = 0.002) with mean survival time of 16.36 months and an SD of 4.15 and 13.95 months and an SD of 5.88 for the V-prep + RMGIC and the conventional method, respectively (Table 5).

**Table 5.** Mean for survival time in months of the different products (N = 200).

|  | Composite | V-Prep + RMGIC | *p*-Value |
|---|---|---|---|
|  | Mean ± SD | Mean ± SD |  |
| **Mean survival time** | 13.95 ± 5.88 | 16.36 ± 4.15 | 0.002 |

When per-maxillary analyses were performed, a statistically significant difference between the mean survival time for the different bonding methods was identified at the level of the mandible (Table 6), with a significantly longer survival time for the V-prep + RMGIC compared to the conventional method (15.52 ± 5.10 vs. 12.50 ± 6.14, *p* = 0.009). No significant difference was found in terms of survival time between the different bonding methods at the upper jaw (*p* > 0.05).

**Table 6.** Mean for survival time of the bracket in months for the different bonding techniques and jaws (N = 200).

| Mean Survival Time | Composite | V-Prep + RMGIC | *p*-Value |
|---|---|---|---|
|  | Mean ± SD | Mean ± SD |  |
| **Upper jaw** | 15.79 ± 5.02 | 17.31 ± 2.44 | 0.104 |
| **Lower jaw** | 12.50 ± 6.14 | 15.52 ± 5.10 | 0.009 |

*3.3. End-of-Treatment Comparison*

Pre-treatment and post-treatment pictures were examined and compared (Figure 1). No color differences were noted between the teeth of different bonding technique groups within the same patient. No white spot lesions were noted on the 25 patients within the 200 bonded premolars in both bonding techniques.

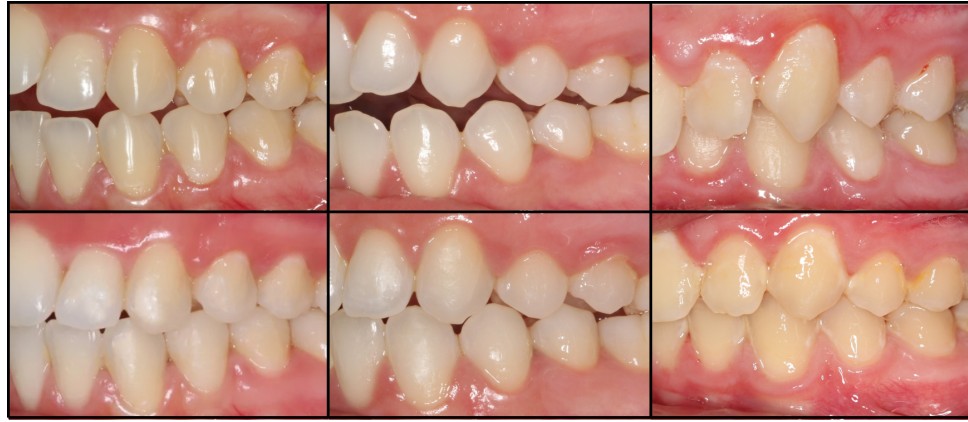

**Figure 1.** Pre-treatment and post-treatment pictures showing premolar changes after eighteen months.

**4. Discussion**

The few data from previous studies of a systematic review regarding the survival rate of different orthodontic bracket bonding techniques were not enough to draw clear conclusions [41]. Multiple variations were not taken into consideration: the length of the monitoring period, the bonding procedure, the light cure time, the initial inclusion criteria of the patients and the study design, etc. This prospective randomized clinical trial showed better bonding failure and survival rates for the V-prep followed by RMGIC over the conventional bonding technique. The study used a single-blind design, involving a within-subject comparison of two bonding systems, with each subject randomly allocated two bonding systems for each premolar of each side of the mouth. As each patient is the unit of assessment for both bonding systems, it is more correct to report the mean bond

failure rate per patient rather than the overall bond failure rate, which may obscure the true nature of the data [24]. Bracket failure may occur in relation to the patient behavior such as type of alimentation, oral hygiene and bad habits, and especially on younger patients (<18 years old) [29]. Thus, this study included only one type of patient, the young patients (12–15 years old) with a class I or mild class II malocclusion to receive real data of the survival rate on patients with higher risk of bonding failure. In this study, only upper and lower premolars were included to minimize the bias of differences in tooth shape and position. Clear protocol for the bonding procedure was applied to enhance the reproduction of this technique for future studies. When a bond failure occurred, a new bracket was bonded after cleaning the tooth surface using magnifiers and following protocols described in a systematic review [34,36]. Crowding and overbite range were limited to mild and average in this study to eliminate a major risk factor [28].

The 37% orthophosphoric acid etch causes microporosities and results in a bonding mechanical retention mechanism via the penetration of the resin tags into the microporous substrate [42,43]. However, the formation of a pellicle coming from the saliva on the enamel surface acts as an acid-resistant coating for teeth [44]. The use of a deproteinization agent as the hypochlorite before applying the acid etch can increase its effect [45]. Thus, the V-prep is a combination of hypochlorite and acid etch giving a better action in the preparation for the mechanical retention part of the RMGIC [16], which was in accordance with the results of this study regarding the bonding failure rates. The increased bond strength of orthodontic brackets means less incidence of debonding during the orthodontic treatment phase but also means penetration of resin tags within enamel that will remain within the tooth enamel structure causing discoloration of the enamel [43]. However, the RMGIC also has a chemical part retention provided from the glass-ionomer cement component which was insufficient without the improvement of the mechanical retention part [14,15]. The combination of the chemical and the increased mechanical retentions improved the bond strength without excessively increasing the penetration of resin tags and causing damage or discoloration to the enamel, as confirmed by the adhesive remnant index in a previous study [16].

Different types of adhesive agents can affect the enamel demineralization for orthodontic brackets bonding. Conventional bonding techniques and self-etch primer adhesives showed significant higher enamel surface demineralization when compared to the RMGIC bonding material in orthodontics [46]. The survival rate observed in this study can encourage the use of the RMGIC as a bonding material to enhance the release of sodium fluoride and inhibit the enamel demineralization during the treatment [47].

Many previous randomized clinical trials investigated the survival rate for six to eighteen months of bracket bonding using two different techniques such as the conventional composite resin Transbond XT (3M Unitek, Monrovia, CA, USA) and the Transbond Plus SEP (3M Unitek, Monrovia, CA, USA) [23,25,26]. Very few studies used the RMGIC as one of the compared bonding techniques in vivo [11,27]. In the latter case, poor results were recorded when comparing the RMGIC to the conventional bonding technique. In this study, the maximum length of monitoring of eighteen months was adopted and a preparation product with reliable in vitro and cytotoxicity results was applied before bonding with RMGIC [16,17].

The bonding failure rate of this study was in accordance with other studies ranging from 2.7 to 23 per cent [40,48–51]. Most of them occurred in the first six months of treatment also in accordance with other studies [23,52,53]. Bracket failure in the first phase of treatment can be caused by the alignment and levelling of crowded and rotated teeth with an increased pressure on weak points. Moreover, the patient's first-time experience may be difficult to manage and commit in the beginning. Other findings, such as bonding failure percentage of the SEP or the conventional separate etch and primer systems, may not be comparable to this study due to the number of operators, different bonding techniques used in the trial, different ages of patients and the inclusion of canines and incisors in the sample [25].

The bonding failure rate of the V-prep and RMGIC combination was lower than the conventional bonding technique, which was not in accordance with other studies that included the RMGIC as bonding agent [11,14]. The use of the RMGIC without any preparation product was compared to the use of the composite resin (Transbond XT, 3M Unitek, Monrovia, CA, USA) with a previous acid-etch preparation. The V-prep cleaned the surface from the organic layer and enhanced the chemical and mechanical bonding of the RMGIC [16]. The isolation problems in vivo can also lead to decreased bond strength of the hydrophobic composite resin [54]. Dry surface, to maximize the composite resin bond efficiency, is not achieved without the use of a rubber dam [55]. RMGIC has more hydrophilic properties that can explain the better results in vivo if the enamel surface is well prepared before its use [56].

The lower jaw showed significantly higher bonding failure rates and lower survival time rates when compared to the upper jaw for both bonding products. Those results were in accordance with other studies [23–29]. The pressure on the lower jaw is higher with the occlusal force direction on the buccal side of lower premolars. The survival rate of the V-prep and RMGIC was significantly higher than the composite resin with a mean value of 16.36 months (1.36 year) which makes the combination an alternative to the conventional technique. The bonding failure between the first and the second premolars on the same jaw and with the same technique did not show a significant difference in this study. These results are in accordance with failure incidence studies according to the tooth type [29,30].

The V-prep and RMGIC, with the potential release of fluoride, were demonstrated to be an effective alternative to the conventional orthodontic bracket bonding technique. In vitro SBS and cytotoxicity comparison to the 37% orthophosphoric acid etch were tested. This clinical trial considered the bracket bond failure and survival rate, the debonding and the rebonding. However, one of the limitations encountered during the trial was the manipulation of the RMGIC (Fuji Ortho LC, GC corporation, Tokyo, Japan) when attempting to bond multiple orthodontic brackets using the same capsule. Increasing the time of auto polymerization may allow the orthodontist to bond a full arch with one capsule of Fuji Ortho LC (GC corporation, Tokyo, Japan) and then start the polymerization by light cure whenever the operator is ready. Thus, it would be interesting to investigate the overall bonding time that may be affected when compared to the conventional bonding technique. Another limitation was the necessity of using a mixer machine to prepare the capsule of the Fuji Ortho LC (GC corporation, Tokyo, Japan) while the conventional technique does not need an extra equipment.

## 5. Conclusions

This eighteen-month randomized clinical trial concluded that the bonding with V-prep and RMGIC can be used as an alternative for the conventional orthodontic bracket bonding technique with a higher bracket survival rate for the V-prep and RMGIC combination. The lower premolars showed higher bonding failure compared to the upper premolars for both bonding techniques. The V-prep prepared the surface for a better adhesion of the RMGIC in vivo enhancing the dispersion of the sodium fluoride and protecting the enamel surface from demineralization.

**Author Contributions:** Conceptualization, V.G., J.G. and E.K.; methodology, L.H., J.G. and E.K.; software, V.G.; validation, V.G., J.G., E.K., M.A., L.H. and T.B.S.; formal analysis, V.G. and J.G.; investigation, V.G., E.K., L.H., M.A. and T.B.S.; resources, V.G., E.K., J.G., L.H. and T.B.S.; data curation, V.G. and E.K.; writing—original draft preparation, V.G., E.K. and J.G.; writing—review and editing, V.G., L.H. and T.B.S.; visualization, V.G., J.G., L.H., T.B.S. and E.K.; supervision, E.K., J.G. and L.H.; project administration, V.G. and E.K. All authors have read and agreed to the published version of the manuscript.

**Funding:** This research received no external funding.

**Institutional Review Board Statement:** The study was conducted in accordance with the Declaration of Helsinki, and approved by the ethical committee of Saint-Joseph University of Beirut (USJ-2020-010).

**Informed Consent Statement:** Informed consent was obtained from all subjects involved in the study.

**Data Availability Statement:** The data that support the findings of this study are available from the corresponding author upon reasonable request.

**Acknowledgments:** Authors would like to acknowledge the Saint-Joseph University of Beirut, Lebanon.

**Conflicts of Interest:** The authors declare no conflict of interest.

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
