# Peer review of "Eighteen-Month Orthodontic Bracket Survival Rate with the Conventional Bonding Technique versus RMGIC and V-Prep: A Split-Mouth RCT"

_coatings, doi:10.3390/coatings13081447_

Round 1

Reviewer 1 Report

Dear Authors,

I have been invited to review your work entitled “Eighteen-month Orthodontic Bracket Survival Rate with the Conventional Bonding Technique versus RMGIC and V-prep: 3 A Split-mouth RCT”. I believe it is a work of concern however, there are some major issues that deserve revisions for acceptance. Please, provide a point-by-point response, highlighting the corrections with a different color mark for each reviewer.

Keywords

Please, add more keywords, you have up to 10 keywords as a limit.

Introduction

  • Please, remove the commercial name. The Introduction should briefly address the topic of the study, and comparisons should be reserved for the Discussion section. 

Materials and methods

·       CONSORT guidelines should be followed to format the manuscript.

·       Sample size calculation is required. By the way, an equal number of patients and brackets should be reached, otherwise an evident bias selection is provided. Please, add more brackets to V-prep + RMGIC group or reduce Composite brackets.  

·       Months and years of beginning and ending of the study are required. 

·       Demographic data of the participants are required. 

Discussion

·       Did the limitation of using RMGIC affect the bonding time? It would have been interesting to evaluate if the chairside time of boding could be affected in someway.

Thank you for the effort.

Minor English revision is required. 

Author Response

Reviewer 1:

Dear Authors,

I have been invited to review your work entitled “Eighteen-month Orthodontic Bracket Survival Rate with the Conventional Bonding Technique versus RMGIC and V-prep: 3 A Split-mouth RCT”. I believe it is a work of concern however, there are some major issues that deserve revisions for acceptance. Please, provide a point-by-point response, highlighting the corrections with a different color mark for each reviewer.

Response: We would like to thank the reviewer 1 for his/her careful reading and constructive comments which gave us the opportunity to improve the quality of our manuscript. We revised the manuscript according to the reviewer’s requests. You will find the point-by-point response below. By comparison with the first version new data are now indicated in yellow.

Reviewer 1:

Keywords

Please, add more keywords, you have up to 10 keywords as a limit.

Response: Keywords were added and highlighted

Reviewer 1:

Introduction

  • Please, remove the commercial name. The Introduction should briefly address the topic of the study, and comparisons should be reserved for the Discussion section.

Response: The only correction we were not able to make was to reduce the introduction and to remove the commercial names because all the studies in the field that were used in the references are referring to a single product with a single commercial name. Different products with the same composition can have big differences in the application process. Reducing the introduction may lead the reader to a lack of data concerning the purpose of using a new product as the V-prep to enhance the efficiency of the RMGIC (pros of the RMGIC, source of the V-prep, why finding an alternative to the conventional technique, etc.)

Reviewer 1:

Materials and methods

  • CONSORT guidelines should be followed to format the manuscript.
  • Sample size calculation is required. By the way, an equal number of patients and brackets should be reached, otherwise an evident bias selection is provided. Please, add more brackets to V-prep + RMGIC group or reduce Composite brackets.  
  • Months and years of beginning and ending of the study are required. 
  • Demographic data of the participants are required. 

Response:

  • CONSORT guidelines were all followed upon your request and the format has been modified accordingly.
  • The sample size calculation has been added. The equal number of bonded brackets was 100 for each group (100x2=200) but the sample size was 239 brackets because the rebonded brackets (39) were also added to the total sample size. However, Reviewer 3 asked to remove the rebonded brackets from the sample to reduce biases. Explanation was logic and the statistics were done again. This matter was clarified in the manuscript and highlighted in yellow and blue.
  • Months and years were also added and highlighted with the add of the clinicaltrial.gov approval number ID
  • Age, sex and status were added in the description of the participants

Reviewer 1:

Discussion

  • Did the limitation of using RMGIC affect the bonding time? It would have been interesting to evaluate if the chairside time of boding could be affected in someway.

Response: We couldn’t really know if it affected an overall time because we were only bonding 4 of 8 premolars with each group and not the total number of teeth in the maxillary or the mandible. However, it will be very interesting to evaluate the chair time in the next study. This matter was noted and highlighted in the manuscript.

Reviewer 2 Report

The authors are requested to add any data regarding the development of any white spot lesions around the bracket during the period tested.  
photographs for the teeth after debonding  should be provided.  
The increased bond strength of orthodontic brackets means less incidence of debonding during the orthodontic treatment phase but also means penetration of resin tags within enamel that will remain within the tooth enamel structure causing discolouration of the enamel.  Please discuss these points and please take reference from the following article.  
Orthodontic Bracket Bonding Using Self-adhesive Cement to Facilitate Bracket Debonding

The English language should be improved and all abbreviations should be omitted completely in the whole manuscript.  The abbreviations are very confusing in the paper.  

Author Response

Reviewer 2:

Response: We would like to thank the reviewer 1 for his/her careful reading and constructive comments which gave us the opportunity to improve the quality of our manuscript. We revised the manuscript according to the reviewer’s requests. You will find the point-by-point response below. By comparison with the first version new data are now indicated in green.

Reviewer 2:

The authors are requested to add any data regarding the development of any white spot lesions around the bracket during the period tested.  
photographs for the teeth after debonding  should be provided.

Response: Pictures are systematically taken before and after each treatment. The authors went back to the data and did the comparison. The data were added and highlighted in the material and methods under a new paragraph titled bracket debonding, data were also added and highlighted in the results and examples of pictures before and after the treatment were added in the figure 2.

Reviewer 2:

The increased bond strength of orthodontic brackets means less incidence of debonding during the orthodontic treatment phase but also means penetration of resin tags within enamel that will remain within the tooth enamel structure causing discolouration of the enamel.  Please discuss these points and please take reference from the following article.  
Orthodontic Bracket Bonding Using Self-adhesive Cement to Facilitate Bracket Debonding

Response: The conventional bonding technique uses the mechanical retention as a main bond strength to the composite and let the increased penetration of the resin tags damage the enamel with an Adhesive remnant index ARI higher on the tooth than on the bracket after the debonding. The main advantage of the RMGIC is the glass ionomer cement component which has a chemical retention higher than the composite with a better ARI for the tooth compared to conventional composite. However, this chemical retention was not sufficient to become an alternative. Enhancing the mechanical low part retention of the other component which the resin can become a good solution when added to the chemical part without increasing the harm on the tooth enamel.

Thank you for bringing up this matter. The points you asked for were discussed and highlighted and the article was mentioned as reference number 43 in the discussion.

Reviewer 2:

The English language should be improved and all abbreviations should be omitted completely in the whole manuscript.  The abbreviations are very confusing in the paper.

Response: RCT abbreviation was removed and replaced (highlighted) with randomized clinical trial. The remaining abbreviations are only RMGIC which is the common name and not only the abbreviation of the resin modified glass ionomer and the V-prep which is not an abbreviation but the name of the new product under testing.

Reviewer 3 Report

This study demonstrated that V-prep + RMGIC Fuji Ortho LC (GIC) is more effective than composite resin (RE) for bracket bonding. Recently, new materials have been developed and their evaluation is important. And studies with low bias will be cited as references for guidelines and systematic reviews. In this regard, there are several concerns with this paper.

1. risk of bias (Rob) should be low. Therefore, it should be clearly indicated as much as possible based on the Rob2 tool (https://www.riskofbias.info/welcome/rob-2-0-tool).

1.1. statement of whether the operator (evaluator) of the periodic examination knows which is the GIC or RE.

1.2. statement that the same treatment was performed on the left and right side at the time of the periodic examinations.

1.3. a statement that no dropouts exist after 18 months.

2. Subjects are 25 first and second premolars, so the total number of teeth is 200. What are the 39 rebonded brackets added in Results paragraph? It is not described in the Materials and Methods paragraph. If the teeth that failed once (N=40) are counted again, they should be excluded because of biases such as more mandibles, more N numbers in the group with more failures, different follow-up periods, etc.

3. Even with the split-mouse method, parameters that differ between groups (129 vs. 110) should be compared. Failure rate may be higher in the mandible compared to the maxilla. The failure rate may be higher for the second premolar, which is more posterior than the first premolar. The tooth type of both groups should be revealed and the absence of bias should be shown.

4. Table 3 shows that the GIC group has 52 teeth in the upper jaw and 58 teeth in the lower jaw, while the RE group has 57 teeth in the upper jaw and 72 teeth in the lower jaw. The RE group clearly has more mandibular. As mentioned previously, the mandible may have a high failure rate. Therefore, the only results with low bias in this study are Table 3 and 5. A high bias occurs when comparing the whole (Tables 2, 4 and Figure 1). Exclusions of the subject teeth should be considered to avoid bias between the respective groups.

Author Response

Reviewer 3:

This study demonstrated that V-prep + RMGIC Fuji Ortho LC (GIC) is more effective than composite resin (RE) for bracket bonding. Recently, new materials have been developed and their evaluation is important. And studies with low bias will be cited as references for guidelines and systematic reviews. In this regard, there are several concerns with this paper.

Response: We would like to thank the reviewer 3 for his/her careful reading and constructive comments which gave us the opportunity to improve the quality of our manuscript. We revised the manuscript according to the reviewer’s requests. You will find the point-by-point response below. By comparison with the first version new data are now indicated in blue.

Reviewer 3:

  1. risk of bias (Rob) should be low. Therefore, it should be clearly indicated as much as possible based on the Rob2 tool (https://www.riskofbias.info/welcome/rob-2-0-tool).

1.1. statement of whether the operator (evaluator) of the periodic examination knows which is the GIC or RE.

1.2. statement that the same treatment was performed on the left and right side at the time of the periodic examinations.

1.3. a statement that no dropouts exist after 18 months.

Response: 1.1. The operator did not know which is GIC or RE during the periodic examination. Statement was noted and highlighted. 1.2. Statement that the same treatment was performed was noted and highlighted. 1.3. Statement that no dropouts was noted and highlighted.

Reviewer 3:

  1. Subjects are 25 first and second premolars, so the total number of teeth is 200. What are the 39 rebonded brackets added in Results paragraph? It is not described in the Materials and Methods paragraph. If the teeth that failed once (N=40) are counted again, they should be excluded because of biases such as more mandibles, more N numbers in the group with more failures, different follow-up periods, etc.

Response: Yes, the rebonded teeth were also added to the sample size before the correction. Upon your interesting remark and to reduce the biases, rebonded teeth were excluded and statistics were done again with N=200 and the total debonded brackets number became 34 instead of 39. The matter was written and highlighted in the methods and in the results.

Reviewer 3:

  1. Even with the split-mouse method, parameters that differ between groups (129 vs. 110) should be compared. Failure rate may be higher in the mandible compared to the maxilla. The failure rate may be higher for the second premolar, which is more posterior than the first premolar. The tooth type of both groups should be revealed and the absence of bias should be shown.

Response: This study is not only a conventional split-mouth study. It is a cross-quadrant split-mouth study meaning that even on the same side and on the same jaw we have different types of techniques to reduce the biases. However, after your comment, we went back through the data and added the table 4 in the manuscript with a comparison of the tooth type as it would be interesting to see if there was a difference. No statistical differences were found and the results were noted and discussed in the discussion (highlighted). These findings were in accordance with other mentioned studies that found a statistical difference between the premolars and the anterior teeth on the bonding failure but did not find a significant difference between the first and the second premolars.

Reviewer 3:

  1. Table 3 shows that the GIC group has 52 teeth in the upper jaw and 58 teeth in the lower jaw, while the RE group has 57 teeth in the upper jaw and 72 teeth in the lower jaw. The RE group clearly has more mandibular. As mentioned previously, the mandible may have a high failure rate. Therefore, the only results with low bias in this study are Table 3 and 5. A high bias occurs when comparing the whole (Tables 2, 4 and Figure 1). Exclusions of the subject teeth should be considered to avoid bias between the respective groups.

Response: Figure 1 was deleted and the exclusion of the rebonded teeth was already considered and done as mentioned before. Table 4 was added to reduce the biases and compare specific parameters together.

Round 2

Reviewer 1 Report

Dear Authors, 

The modifications performed make the manuscript suitable for publication.

Thank your for your hard work.

Minor English revision is required. 

Reviewer 3 Report

The article has been carefully revised. Bias would have been lowered and quality improved.